# The Effects of Invertibility on the Representational Complexity of Encoders in Variational Autoencoders

## Abstract

Training and using modern neural-network based latent-variable generative models (like Variational Autoencoders) often require simultaneously training a generative direction along with an inferential (encoding) direction, which approximates the posterior distribution over the latent variables. Thus, the question arises: how complex does the inferential model need to be, in order to be able to accurately model the posterior distribution of a given generative model? In this paper, we identify an important property of the generative map impacting the required size of the encoder. We show that if the generative map is "strongly invertible" (in a sense we suitably formalize), the inferential model need not be much more complex. Conversely, we prove that there exist non-invertible generative maps, for which the encoding direction needs to be exponentially larger (under standard assumptions in computational complexity). Importantly, we do not require the generative model to be layerwise invertible, which a lot of the related literature assumes and isn't satisfied by many architectures used in practice (e.g. convolution and pooling based networks). Thus, we provide theoretical support for the empirical wisdom that learning deep generative models is harder when data lies on a low-dimensional manifold.

## 1. Introduction

Many modern generative models of choice (e.g. Generative Adversarial Networks (Goodfellow et al., 2014), Variational Autoencoders (Kingma and Welling, 2013)) are modeled as non-linear, possibly stochastic transformations of a simple latent distribution (e.g. a standard Gaussian). A particularly common task is modeling the *inferential (encoder) direction*: that is, modeling the posterior distribution on the latents $z$ given an observable sample $x$. Such a task is useful both at *train time* and at *test time*. At train time, fitting generative models like variational autoencoders via maximum likelihood often relies on variational methods, which require the joint training of a *generative model (i.e. generator/decoder)*, as well as an *inference model (i.e. encoder)* which models the posterior distribution of the latent given the observables. At test time, the posterior distribution very often has some practical use, e.g. useful, potentially interpretable feature embeddings for data (Berthelot et al., 2018), "intervening" on the latent space to change the sample in some targeted manner (Shen et al., 2020), etc. As such, the question of the "complexity" of the inference model (i.e. number of parameters to represent it using a neural network-based encoder) as a function of the "complexity" of the forward model is of paramount importance:

**Question:** *How should we choose the architecture of the inference (encoder) model relative to the architecture of the generative (decoder) model during training?*

For instance, when is the backward model not much more complex, so that training in this manner is not computationally prohibitive? Such a question is also pertinent from a purely scientific perspective, as it asks:

**Question:** *Given a generative model for data, when is inference (much) harder than generation?*

In this paper we identify an important aspect of the generative direction governing the complexity of the inference direction for *variational autoencoders*: a notion of approximate *bijectivity/invertibility* of the mean of the generative direction. We prove that under this assumption, the complexity of the inference direction is not much greater than the complexity of the generative direction. Conversely, without this assumption, under standard computational complexity conjectures from cryptography, we can exhibit instances where the inference direction has to be much more complex.

On the mathematical level, our techniques involve a neural simulation of a Langevin random walk to sample from the posterior of the latent variables. We show that the walk converges fast when started from an appropriate initial point

---

[1]Anonymous Institution, Anonymous City, Anonymous Region, Anonymous Country. Correspondence to: Anonymous Author <anon.email@domain.com>.

Preliminary work. Under review by INNF+ 2021. Do not distribute.

– which we can compute using gradient descent (again, simulated via a neural network). On the lower bound side, we provide a reduction from the existence of *one-way Boolean permutations* in computational complexity: that is, permutations that are easy to calculate, but hard to invert. We show that the existence of a small encoder for non-invertible generators would allow us to design an invertor for *any* Boolean permutation, thus violating the existence a one-way permutation. This is the first time such ideas have been applied to generative models.

Our results can be seen as corroborating empirical observations that learning deep generative models more generally is harder when data lies on a low-dimensional manifold (Dai and Wipf, 2019; Arjovsky et al., 2017).

## 2. Our Results

The Variational Autoencoder (VAE) (Kingma and Welling, 2013) is one of the most commonly used paradigms in generative models. It's trained by fitting a *generator* which maps latent variables $z$ to observables $x$, denoted by $p_\theta(x|z)$, as well as an *encoder* which maps the observables to the latent space, denoted by $q_\phi(z|x)$. Here $\phi$ and $\theta$ are the encoder parameters and generator parameters respectively. Given $n$ training samples $\{x^{(i)}\}_{i=1}^n$, the VAE objective is given by

$$\max_{\phi,\theta} \frac{1}{n} \sum_{i=1}^n \mathbb{E}_{z \sim q_\phi(.|x^{(i)})} \left[ \log p_\theta(x^{(i)}|z) \right]$$
$$- KL \left( q_\phi(z|x^{(i)}) || p(z) \right)$$

where $p(z)$ is typically chosen to be a standard Gaussian. This loss can be viewed as a variational relaxation of the maximum likelihood objective, where the encoder $q_\phi$, in the limit of infinite representational power, is intended to model the posterior distribution $p_\theta(z|x^{(i)})$ over the latent variables $z$.

**Setup:** We will consider a setting in which the data distribution itself is given by some ground-truth generator $G : \mathbb{R}^{d_l} \to \mathbb{R}^{d_o}$, and ask how complex (in terms of number of parameters) the encoder needs to be (as a function of the number of parameters of $G$), s.t. it approximates the posterior distribution $p(z|x)$ of the generator.

We will consider two standard probabilistic models for the generator/encoder respectively.

**Definition 1** (Latent Gaussian). A *latent Gaussian* is the conditional distribution given by a stochastic pushforward of the standard gaussian. That is, for latent variable $z \in \mathbb{R}^{d_l}$ and observable $x \in \mathbb{R}^{d_o}$, for a neural network $G : \mathbb{R}^{d_l} \to \mathbb{R}^{d_o}$ and noise parameter $\beta^2$; the distribution $p(x|z) = \mathcal{N}(G(z), \beta^2 I_{d_o})$ is a latent Gaussian when $p(z) = \mathcal{N}(0, I_{d_l})$.

In other words, a sample from this distribution can be generated by sampling independently $z \sim \mathcal{N}(0, I)$ and $\xi \sim \mathcal{N}(0, \beta^2 I)$ and outputting $x = G(z) + \xi$. This is a standard neural parametrization of a generator with (scaled) identity covariance matrix, a fairly common choice in practical implementations of VAEs (Kingma and Welling, 2013; Dai and Wipf, 2019).

We will also define a probabilistic model which is a *composition* of latent Gaussians (i.e. consists of multiple *stochastic* layers), which is also common, particularly when modeling encoders in VAEs, as they can model potentially non-Gaussian posteriors (Burda et al., 2015; Rezende et al., 2014):

**Definition 2** (Deep Latent Gaussian). A *deep latent Gaussian* is the conditional distribution given by a sequence of stochastic pushforwards of *any* density. That is, for observable $z_0 \in \mathbb{R}^{d_0}$ and latent variables $\{z_i \in \mathbb{R}^{d_i}\}_{i=1}^L$, for neural networks $\{G_i : \mathbb{R}^{d_{i-1}} \to \mathbb{R}^{d_i}\}_{i=1}^L$ and noise parameters $\{\beta_i^2\}_{i=1}^L$, the conditional distribution $p(z_L|z_0)$ is a deep latent Gaussian when $p(z_i|z_{i-1}) = \mathcal{N}(G_i(z_{i-1}), \beta_i^2 I_{d_i}), \forall i \in [L]$ and $p(z_0)$ is any valid density.

In other words, a deep latent Gaussian is a distribution, which can be sampled by ancestral sampling, one layer at a time. Note that this class of distributions is convenient as a choice for an encoder in a VAE, since compositions are amenable to the reparametrization trick of (Kingma and Welling, 2013) – the randomness for each of the layers can be "presampled" and appropriate transformed (Burda et al., 2015; Rezende et al., 2014). Then, we ask the following:

**Question:** If a VAE generator is modeled as a latent Gaussian (that is, $p(x|z) \equiv \mathcal{N}(G(z), \beta^2 I)$), s.t. the corresponding $G$ has at most $N$ parameters, and we wish to approximate the posterior $p(z|x)$ by a deep latent Gaussian s.t. the total size of the networks in it have at most $N'$ parameters, how large must $N'$ be as function of $N$?

We will work in the setting $d_l = d_o = d$, and prove a dichotomy based on the invertibility of $G$: namely, if $G : \mathbb{R}^d \to \mathbb{R}^d$ is bijective, and $\beta \leq \mathcal{O}\left(\frac{1}{d^{1.5}\sqrt{\log d/\epsilon}}\right)$, the posterior $p(z|x)$ can be $\epsilon$-approximated in total variation distance by a deep latent Gaussian of size $N' = \mathcal{O}(N \cdot poly(d, 1/\beta, 1/\epsilon))$. Thus, if the neural network $G$ is invertible, and for a fixed $\epsilon$ and a small-enough variance term $\beta^2$, we can approximate the posterior with a deep latent Gaussian polynomially larger than $G$. On the other hand, if $G$ is not bijective, if one-way-functions exist (a widely believed computational complexity conjecture), we will show there exists a VAE generator $G : \mathbb{R}^d \to \mathbb{R}^d$ of size polynomial in $d$, for which the posterior $p(z|x)$ cannot be approximated in total variation distance for *even an inverse polynomial fraction* of inputs $x$, unless the inferential

network is of size *exponential* in $d$.

## 2.1. Upper bounds for bijective generators

We first lay out the assumptions on the map $G$. The first is a quantitative characterization of bijectivity; and the second requires upper bounds on the derivatives of $G$ upto order 3. We also have a centering assumption. We state these below.

**Assumption 1** (Strong invertibility). *We will assume that the latent and observable spaces have the same dimension (denoted $d$), and $G : \mathbb{R}^d \to \mathbb{R}^d$ is bijective. Moreover, we will assume there exists a positive constant $m > 0$ such that:*

$$\forall z_1, z_2 \in \mathbb{R}^d, \ \ \|G(z_1) - G(z_2)\| \geq m \cdot \|z_1 - z_2\|$$

**Remark 1:** This is a stronger quantitative version of invertibility. Furthermore, the infinitesimal version of this condition (i.e. $\|z_1 - z_2\| \to 0$) implies that the smallest magnitude of the singular values of the Jacobian at any point is lower bounded by $m$, that is $\forall z \in \mathbb{R}^d, \ \min_{i \in [d]} |\sigma_i(J_G(z))| \geq m > 0$. Since $m$ is strictly positive, this in particular means that the Jacobian is full rank everywhere.

**Remark 2:** Note, we do *not* require that $G$ is layerwise invertible (i.e. that the each map from one layer to the next is invertible) – if that is the case, at least in the limit $\beta \to 0$, the existence of an inference decoder of comparable size to $G$ is rather obvious: we simply invert each layer one at a time. This is important, as many architectures based on convolutions perform operations which increase the dimension (i.e. map from a lower to a higher dimensional space), followed by pooling (which decrease the dimension). Nevertheless, it has been observed that these architectures are invertible in practice— (Lipton and Tripathi, 2017) manage to get almost 100% success at inverting an off-the-shelf trained model— thus justifying this assumption.

**Assumption 2** (Smoothness). *There exists a finite positive constant $M > 0$ such that :*

$$\forall z_1, z_2 \in \mathbb{R}^d, \ \ \|G(z_1) - G(z_2)\| \leq M \cdot \|z_1 - z_2\|$$

*Moreover, we will assume that $G$ has continuous partial derivatives up to order 3 at every $z \in \mathbb{R}^d$ and the derivatives are bounded by finite positive constants $M_2$ and $M_3$ as*

$$\forall z \in \mathbb{R}^d, \ \ \left\|\nabla^2 G(z)\right\|_{op} \leq M_2 < \infty,$$

$$\left\|\nabla^3 G(z)\right\|_{op} \leq M_3 < \infty$$

**Remark 3:** This is a benign assumption, stating that the map $G$ is smooth to third order. The infinitesimal version of this means that the largest magnitude of the singular values of the Jacobian at any point is upper bounded by $M$, that is $\forall z \in \mathbb{R}^d, \ \max_{i \in [d]} |\sigma_i(J_G(z))| = \|J_G(z)\|_{op} \leq M < \infty$.

**Remark 4:** A neural network with activation function $\sigma$ will satisfy this assumption when $\sigma : \mathbb{R} \to \mathbb{R}$ is Lipschitz, and $\max_a |\sigma'(a)|$ & $\max_a |\sigma''(a)|$ are finite.

**Assumption 3** (Centering). *The map $G : \mathbb{R}^d \to \mathbb{R}^d$ satisfies $G(0) = 0$.*

**Remark 5:** This assumption is for convenience of stating the bounds — we effectively need the "range" of majority of the samples $x$ under the distribution of the generator. All the results can be easily restated by including a dependence on $\|G(0)\|$.

Our main result is then stated below. Throughout, the $\mathcal{O}(.)$ notation hides dependence on the map constants, namely $m, M, M_2, M_3$. We will denote by $d_{\text{TV}}(p, q)$ the total variation distance between the distributions $p, q$.

**Theorem 1** (Main, invertible generator). *Consider a VAE generator given by a latent Gaussian with noise parameter $\beta^2$ and generator $G : \mathbb{R}^d \to \mathbb{R}^d$ satisfying Assumptions 1 and 2, which has $N$ parameters and a differentiable activation function $\sigma$. Then, for*

$$\beta \leq \mathcal{O}\left(\frac{1}{d^{1.5}\sqrt{\log \frac{d}{\epsilon}}}\right) \quad (1)$$

*there exists a deep latent Gaussian with $N' = \mathcal{O}\left(N \cdot poly(d, \frac{1}{\beta}, \frac{1}{\epsilon})\right)$ parameters and activation functions $\{\sigma, \sigma', \rho\}$, where $\rho(x) = x^2$, such that with probability $1 - \exp(-\mathcal{O}(d))$ over a sample $x$ from the VAE generator, the distribution $q(z|x)$ of the deep latent Gaussian on input $x$ satisfies $d_{TV}(q(z|x), p(z|x)) \leq \epsilon$.*

**Remark 6:** The addition of $\rho$ in the activation functions is for convenience of stating the bound. Using usual techniques in universal approximation it can be simulated using any other smooth activation.

## 2.2. Lower bounds for non-bijective Generators

We now discuss the case when the generative map $G$ is not bijective, showing an instance such that no small encoder corresponding to the posterior exists. The lower bound will be based on a reduction from the existence of one-way functions – a standard complexity assumption in theoretical computer science (more concretely, cryptography). Precisely, we will start with the following form of the one-way-function conjecture:

**Conjecture 1** (Existence of one-way permutations, (Katz and Lindell, 2020)). *There exists a bijection $f : \{-1, 1\}^d \to \{-1, 1\}^d$ computable by a Boolean circuit $\mathcal{C} : \{-1, 1\}^d \to \{-1, 1\}^d$ of size $poly(d)$, but for every $T(d) = poly(d)$ and $\epsilon(d) = \frac{1}{poly(d)}$ and circuit $\mathcal{C}' : \{-1, 1\}^d \to \{-1, 1\}^d$ of size $T(d)$ it holds $\mathbf{Pr}_{z \sim \{\pm 1\}^d}[\mathcal{C}'(\mathcal{C}(z)) = z] \leq \epsilon(d)$.*

In other words, there is a circuit of size polynomial in the input, s.t. for every polynomially sized invertor circuit (the two polynomials need not be the same – the invertor can be much larger, so long as it's polynomial), the invertor circuit succeeds on at most an inverse polynomial fraction of the inputs. Assuming this Conjecture, we show that there exist generators that do not have small encoders that accurately represent the posterior for most points $x$. Namely:

**Theorem 2** (Main, non-invertible generator). *If Conjecture 1 holds, there exists a VAE generator $G : \mathbb{R}^d \to \mathbb{R}^d$ with size $poly(d)$ and activation functions $\{sgn, \min, \max\}$, s.t., for every $\beta = o(1/\sqrt{d})$, every $T(d) = poly(d)$ and every $\epsilon(d) = 1/poly(d)$, any encoder $E$ that can be represented by a deep latent Gaussian with networks that have total number of parameters bounded by $T(d)$, weights bounded by $W$, activation functions that are $L$-Lipschitz, and node outputs bounded by $M$ with probability $1 - \exp(-d)$ over a sample $x$ from $G$ and $L, M, W = o(\exp(poly(d)))$, we have:*

$$\mathbf{Pr}_{x \sim G}\left[d_{TV}(E(z|x), p(z|x)) \leq \frac{1}{10}\right] \leq \epsilon(d)$$

Thus, we show the existence of a generator for which no encoder of polynomial size reasonably approximates the posterior for *even an inverse-polynomial* fraction of the samples $x$ (under the distribution of the generator).

**Remark 7:** The generator $G$, though mapping from $\mathbb{R}^d \to \mathbb{R}^d$ will be highly *non-invertible*. Perhaps counterintuitively, Conjecture 1 applies to bijections—though, the point will be that $G$ will be simulating a Boolean circuit, and in the process will give the same output on many inputs (more precisely, it will only depend on the sign of the inputs, rather than their values).

**Remark 8:** The choice of activation functions $\{sgn, \min, \max\}$ is for convenience of stating the theorem. Using standard universal approximation results, similar results can be stated with other activation functions.

**Remark 9:** The restrictions on the Lipschitzness of the activations, bounds of the weights and node outputs of $E$ are extremely mild – as they are allowed to be potentially exponential in $d$ – considering that even writing down a natural number in binary requires logarithmic number of digits.

## 3. Related Work

On the empirical side, the impact of impoverished variational posteriors in VAEs (in particular, modeling the encoder as a Gaussian) has long been conjectured as one of the (several) reasons for the fuzzy nature of samples in trained VAEs. (Zhao et al., 2017) provide recent evidence towards

this conjecture. Invertibility of generative models in general (VAEs, GANs and normalizing flows), both as it relates to the hardness of fitting the model, and as it relates to the usefulness of having an invertible model, has been studied quite a bit: (Lipton and Tripathi, 2017) show that for off-the-shelf trained GANs, they can invert them with near-100% success rate, despite the model not being encouraged to be invertible during training; (Dai and Wipf, 2019) propose an alternate training algorithm for VAEs that tries to remedy algorithmic problems during training VAEs when data lies on a lower-dimensional manifold; (Behrmann et al., 2020) show that trained normalizing flows, while being by design invertible, are just barely so — the learned models are extremely close to being singular.

On the theoretical side, the most closely relevant work is (Lei et al., 2019). They provide an algorithm for inverting GAN generators *with random weights* and *expanding layers*. Their algorithm is layerwise — that is to say, each of the layers in their networks is invertible, and they invert the layers one at a time. This is distinctly not satisfied by architectures used in practice, which expand and shrink — a typical example are convolutional architectures based on convolutions and pooling. The same paper also shows NP-hardness of inverting a general GAN, but crucially they assume the network $G$ is part of the input (their proof does not work otherwise). Our lower bound can be viewed as a "non-uniform complexity" (i.e. circuit complexity) analogue of this, since we are looking for a small neural network $E$, as opposed to an efficient algorithm; crucially, however $G$ is *not* part of the input (i.e. $G$ can be preprocessed for an unlimited amount of time). (Hand and Voroninski, 2018) provide similar guarantees for inverting GAN generators with random weights that satisfy layerwise invertibility, albeit via non-convex optimization of a certain objective.

## 4. Conclusion

In this paper we initiated the first formal study of the effect of invertibility of the generator on the representational complexity of the encoder in variational autoencoders. We proved a dichotomy: invertible generators give rise to distributions for which the posterior can be approximated by an encoder not much larger than the generator. On the other hand, for non-invertible generators, the corresponding encoder may need to be exponentially larger. Our work is the first to connect the complexity of inference to invertibility, and there are many interesting venues for further work.

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
