# OpenReview forum: "The Effects of Invertibility on the Representational Complexity of Encoders in Variational Autoencoders"
_ICML.cc/2021/Workshop/INNF — INNF+ 2021 poster_

### Official Review · Reviewer_Wjkp · 2021-06-13

**Rating:** Accept
**Confidence:** 3

**Summary:**

This paper studies the effects of invertibility on the representational complexity of encoders.
It connects the complexity of inference to invertibility and provides theoretical insights into why learning deep generative models is harder when data lies on a low-dimensional manifold.


**Justification For Rating:**

1. The paper is well written.
2. The proposed idea is novel. It connects the complexity of inference to invertibility and provides new perspectives for understanding the representational complexity of encoders in variational autoencoders. This new perspective allows for new research directions.
3. It would be great if the proofs of the theorems can also be provided (maybe in appendix).

---

### Official Review · Reviewer_LmLH · 2021-06-13

**Rating:** Borderline Accept
**Confidence:** 1

**Summary:**

The paper presents a theoretical result stating that for smooth invertible decoders calculating the posterior distribution over latent variables conditioned on an observation is possible using a polynomial number of parameters compared to the decoder. However, among non smooth-and-invertible decoders there must exist cases where the number of parameters required for inversion is exponentially larger.

The paper presents a worst-case scenario type of analysis, and it is not clear how relevant these considerations are for practical use. However, the authors adequately reference previous work indicating that the hardness of evaluating the posterior may be common.

**Justification For Rating:**

The ideas presented in the paper are interesting, and well presented. However, an appendix with proofs for the main results should be added. Maybe due to this reviewer's lack of familiarity with cryptographic assumptions, a leap of faith is necessary to believe the main results presented.

The paper could also benefit from empirical results motivating the theoretical analysis.

---

### Official Review · Reviewer_FNUp · 2021-06-13

**Rating:** Borderline Accept
**Confidence:** 2

**Summary:**

This paper presents two results: (i) that when the generator of a VAE is invertible with bounded smallest and largest singular values, and when the observation noise is small enough, there exist a "small" encoder that has a small TV distance between the encoder and the true posterior. (ii) There exist a VAE generator such that no polynomial-sized encoder can represent its posterior for more than an inverse-polynomial percentage of samples.

**Justification For Rating:**

I don't presume to fully understand this work, but the line of inquiry is interesting, and it provides a formal statement for the advantage of using an invertible generator in a VAE. This is certainly relevant to the workshop.

I'd be curious to understand (i) how strong these bounds on beta are (is there really still any marginalization effect?), (ii) how important the choice of a *deep* latent Gaussian encoder is (is an autoregressive encoder necessary?).

---

### Decision · Program_Chairs · 2021-06-14

Accept (poster)